# A Trefoil Knot Polymer Chain Translocates through a Funnel-like Channel: A Multi-Particle Collision Dynamics Study

**DOI:** 10.3390/polym14061164

**Published:** 2022-03-15

**Authors:** Xiaohui Wen, Deyin Wang, Jiajun Tang, Zhiyong Yang

**Affiliations:** 1Department of Physics, Chengdu University of Technology, Chengdu 610059, China; wenxiaohui13@cdut.edu.cn (X.W.); evenwong@stu.cdut.edu.cn (D.W.); tangmr0101@163.com (J.T.); 2Department of Physics, Jiangxi Agricultural University, Nanchang 330045, China

**Keywords:** trefoil knot chain, solvent, multi-particle collision dynamics, funnel-like channel

## Abstract

With combining multi-particle collision dynamics (MPCD) for the solvent and molecular dynamics (MD) for the polymer chains, we have studied the conformation and untying behaviors of a trefoil knot polymer chain translocated through a confined funnel-like channel. For the trefoil knot chain, we found that the untying knot behavior mostly happens during the translocation process, and the translocation behavior of linear chains is also simulated as a comparison. Some characteristics of the trefoil knot chain during translocation process, such as average gyration radius <Rg> and the average end-to-end distances <S> are discussed, and we statistic the scale relations of the translocation time versus the chain length, and that of the chain rigidity. This study may help to understand translocation behaviors of the knotted linear polymer chain in the capillary flow field.

## 1. Introduction

The prevalence of knots in our everyday life, be it in the form of entangled electric cords or hair, suggests that knots are a common occurrence. Indeed, it has been proven that the knotting probability of a chain tends to unity as the chain length approaches infinity, hence, it is inevitable that knots are present on long chains. Knots are encountered not only at the macro-scale, but also at the micro-scale, having been observed in biopolymers, such as DNA and proteins, as well as in synthetic polymers. Due to the special structural characteristics and unknown properties, more and more researchers are interested in the phenomenon of polymer knotting in recent years [1,2,3,4,5,6]. Experimentally, under elongation flow field, Patrick S. Doyle et al. studied the untying of complex knots, motion of knots in DNA and the dynamics of DNA knots during chain relaxation [7,8,9]; they found that the change in knot size during the knot untying process leads to a change in Wi_eff_, which in turn causes a change in chain extension; and observed that knots moved noticeably across the molecule on minute-long timescales until they reached the end and untied. Davide Michieletto et al. have researched the topological patterns in two-dimensional gel electrophoresis of DNA knots [10], they showed that the mobility of DNA knots depends crucially and subtly on the physical properties of the gel and on the presence of dangling ends. In the area of theory and simulation, Mattia Marenda et al. studied the hydrodynamical transport, filtering and sorting ring polymers by knot type with modulated channels [11], the longitudinal mobility of the rings is ideally suited for knot sorting; and this complex interplay of channel geometry, chain length and topology can be rationalized within a simple theoretical framework based on Fick–Jacobs’s diffusive theory, thus the results and the interpretative scheme ought to be useful for designing microfluidic devices with optimal topological sorting capabilities. With the research development of knotted polymer chains, more and more researchers are interested in the problem of how the knotted polymer chain translocates in a confined environment. In addition, in order to make the simulation results more realistic, researchers mimic the confined environment filled with solvent particles.

The fluid properties under confined environments, such as nano or microfluidic devices, have been studied intensively in recent years [12,13,14,15,16,17]. In such system, factors, which include confined environment, polymer chain and solvent particles, will interplay with each other, the emergent properties of the polymer chain are of their combined actions. At present, some researchers have developed some mesoscopic simulation techniques to model the flow field, e.g., the dissipative particle dynamics (DPD) approach [14,18,19] and the multi-particle collision dynamics (MPCD) method [16], specially, more and more researchers have made use of MPCD to simulate the complex fluid because of its superiority [1,12,13,20,21,22]. This method has been carefully studied in reference [16], the fluid is modeled by “particles” whose positions and velocities are continuous variables, while time is discretized. The evolution of the system occurs in two steps: streaming and collision. This method conserves mass, momentum and energy, and the Galilean invariance is critically discussed [22]. The proper account of hydrodynamic interactions (HI) is also implemented.

In this article, MPCD for the solvent is combined with molecular dynamics (MD) for the polymer to study the characteristics of a knotted linear polymer chain (3_1_ trefoil knot) in a funnel-like microfluidic device. We systematically investigate how the factors, which include the confined environment and solvent particles, affect the conformations and dynamic behaviors of the knotted linear polymer.

This article is organized as follows. The model and simulation approach are introduced in Section 2. Section 3 describes the simulation results, mainly discussing the Poiseuille flow field as well as the conformations and dynamics process of the knotted polymer chain translocation. Conclusions are presented in Section 4.

## 2. Model and Simulation Methods

Our initial system is comprised of three parts: (1) a single knotted linear polymer chain consisting of multiple monomers, which is the main interest of this research object; (2) many solvent particles serving as a solution environment; and (3) a periodic sloping micro-channel with fixed geometry size, which illustrates the confinement boundary conditions. The first and second parts are confined in the last part. A periodic condition is applied along the axial direction of the cylinder (*z*-axis). In the simulation, reduced units are used for simplicity.

### 2.1. Dynamics Behaviors of the Knotted Liner Polymer Chain

We consider a trefoil knotted linear semi-flexible polymer chain embedded in an explicit solvent environment. The polymer chain is composed by *N_m_* beads, and the mass of each bead is *M*. All the beads are connected by finitely extensible nonlinear elastic (FENE) springs, whose potential can be described with:(1)$$U_{\mathrm{FENE}}=-\frac{\kappa }{2}R_{0}^{2}\ln\left(1-{\left(\frac{r}{R_{0}}\right)}^{2}\right),r<R_{0}$$
where *κ* is the spring constant of FENE bonds, *r* is the distance between two adjacent monomers and *R*_0_ is the maximum extension of the spring.

The shifted and truncated Lennard-Jones (LJ) potential
(2)$$U_{\mathrm{LJ}}=\left\{\begin{array}{cc} 4\varepsilon \left({\left(\frac{\sigma }{r}\right)}^{12}-{\left(\frac{\sigma }{r}\right)}^{6}\right)-4\varepsilon \left({\left(\frac{\sigma }{r_{c}}\right)}^{12}-{\left(\frac{\sigma }{r_{c}}\right)}^{6}\right), & r\leq r_{c}\\ 0, & r>r_{c} \end{array}\right.$$
is used to account for non-bonded interactions; σ is the size of a bead and ε is the interaction energy. Intermolecular excluded-volume interactions between beads with the indices 1,…, *N_m_* within a polymer are captured by *r_c_* = 2^1/6^σ, which implies exclusively repulsive potential.

The stiffness of the polymer chain is simulated by the bending potential:(3)$$U_{bend}=b\left(1+\cos\theta \right)$$
where *θ* represents the subtended angle between any two consecutive bonds on the chain, and b denotes the bending energy coefficient, which quantifies the stiffness of the chain. b = 0, > 0 and →∞ indicate that the chain is a completely flexible, semi-flexible and rigidly aligned rod, respectively. In this study, we will mainly focus on the semi-flexible polymer.

The monomer dynamics are governed by Newton’s equations of motion. With the time step is *h_p_*, the coordinates and velocities are updated by the velocity-Verlet algorithm.

### 2.2. Multi-Particle Collision Dynamics for the Fluid Simulation

In order to create a smooth laminar flow field, a funnel shaped segment is used to simulate the confined environment. The solvent is confined in the funnel-like cylindrical channel of length *L*, which consists of wide diameter *D* and narrow diameter *d* along the *z*-axis, as shown in Figure 1. Periodic boundary conditions along the *z*-axis direction are applied. Flow is induced by a gravitational force acting on every fluid particle along the direction of the channel axis (i.e., *z*-axis), which corresponds to a pressure-driven flow, denoted as Poiseuille flow field. Obviously, the system conducts a non-equilibrium process.

The MPC solvent is composed of *N_s_* point particles with mass *m*. The solvent particles interact with each other by a stochastic process, which consists of two steps: streaming and collision. In the streaming step, the particles move ballistically and their positions are updated as
(4)$$r_{i\beta }(t+h)=r_{i\beta }(t)+hv_{i\beta }(t),\;\beta =x,y$$
(5)$$r_{iz}(t+h)=r_{iz}(t)+hv_{iz}(t)+\frac{1}{2}gh^{2}$$
(6)$${\hat{v}}_{iz}(t+h)=v_{iz}(t)+gh$$
where, ***r**_i_* and ***v****_i_* are the position and velocity of the *i*-th particle, respectively. *h* is the time between collisions, and we set *γ* = *gh* for simplicity in the simulation process. ${\hat{v}}_{iz}(t+h)$ is the velocity along the *z*-axis after streaming.

In the collision step, the particles are sorted into cubic cells with side *a*. Then, in each cell, the relative velocities to the center-of-mass velocity after streaming (i.e., ${\hat{v}}_{cm}(t+h)$) of the particles, including monomers and solvents, are rotated around a randomly orientated axis with a fixed angle *α* [23].
(7)$$v_{i}(t+h)={\hat{v}}_{cm}(t)+R(\alpha )\left({\hat{v}}_{i}(t+h)-{\hat{v}}_{cm}(t+h)\right)$$
where ${\hat{v}}_{i}(t+h)=v_{i}(t)$ is the velocity after streaming, and before collision. *R*(*α*) is a rotation matrix, in which the rotation axis is chosen randomly for every collision cell and time step.

The solvent-monomer coupling is introduced by including the monomers in the collision step, in which the velocity of the center-of-mass of a cell is calculated as [24]
(8)$$v_{cm}(t)=\frac{\sum_{i=1}^{N_{s}^{c}}mv_{i}(t)+\sum_{k=1}^{N_{m}^{c}}Mv_{k}(t)}{mN_{s}^{c}+MN_{m}^{c}}$$
where $N_{s}^{c}$ and $N_{m}^{c}$ are the number of solvent particles and monomers in the cell, respectively, and the entire average particle number in a collision cell is set to $<N^{c}>$.

Periodic boundary conditions are applied in the *z*-direction, while *x*- and *y*-directions are confined by cylindrical channel. When a random shift of the cell lattice before streaming is required to guarantee Galilean invariance, partially occupied boundary cells are unavoidable. Then, the simple bounce-back rule, i.e., the velocscities of particles which hit the wall are inverted after the collision, fails to guarantee no-slip boundary conditions in the case of partially filled cells. To reduce fluid slip at the walls, the virtual particles approach is adopted as proposed in reference [22]. That is to say, for all the cells of the channel which are cut by walls and therefore have a number of particles smaller than the average number of the bulk cells, we fill the “wall” part of the cells with virtual particles in order to make the effective density of real plus virtual particles equal to the average density. The velocities of the virtual particles are drawn from a Maxwell–Boltzmann distribution of zero average velocity. In one collision step, the relative velocities with respect to center-of-mass velocity of the particles within this cell are rotated, the process is similar to the bulk cells. Specially, the virtual particles are static for simplicity, but their velocities are reset at each step. This boundary condition is called residual slip.

### 2.3. Parameters

In Figure 1, we use *D* = 10.0*a*, *d* = 2.0*a* to mimic the funnel-like cylinder channel with its entire length of *L* = 100*a*, *l* = 50.0*a* and *l*_0_ = 20.0*a*. The periodic and the flow direction are along the *z*-axis direction. The independent parameters of the MPC method are denoted in reduced units: *m* = 1.0, *k_B_T* = 1, *a* = 1 and *h* = 0.1*τ*, where *τ* = (*ma^2^*/*k_B_T*)^1/2^ is the time unit, α = 130° and $<N^{c}>=3$ simultaneously. These values are selected with the purpose of tuning the characteristics of the MPC fluid in a regime where collisional effects dominate the propagation or kinematic effects. In addition, we adopt *N_m_* = 50, 100, 150, 200 and 250 monomers for the linear polymer chains and that of the trefoil knot polymer chains. Every polymer monomer has a diameter σ = 1, and monomer mass *M* = 5*m =* 5.0 and *κ* = 30 (*k_B_T*/*a*^2^); various stiffness parameters—b = 0, 10, 100, 1000 and 10,000 are considered. Moreover, between two collision steps, 50 MD simulation steps are performed for the monomers, i.e., the MD time step is *h_p_* = *h*/50 = 0.002*τ*. At the beginning of the simulation, the linear or knotted linear polymer chain was randomly placed in the funnel-like cylinder channel and equilibrated without any external flow field. After turning on the external Poiseuille flow, the production process conducts for 2 × 10^6^, 3 × 10^6^, 4 × 10^6^, 5 × 10^6^ and 6 × 10^6^ time steps for different *N_m_*, respectively. The statistical data mentioned below are the averages of five samples.

## 3. Simulation Results and Discussions

### 3.1. The Poiseuille Flow

Figure 2a illustrates the cross section of the Poiseuille flow of *g* = 0.01, where Figure 2a indicts the flow field of the funnel-like channel, in order for simplicity, we show only the partial flow field. One can see that the flow rate of the narrow channel is larger than that of the wide part, and the flow field exhibits parabolic profiles at the wide cylinders, which vividly portray the Poiseuille flow. In order to quantitatively mimic the parabolic profile, we calculate the cut-cross flow rates along the radius direction with *z* = −40, −30, −20, −15, −10, 5, 15, 25, 35 and 45, respectively, which is shown in Figure 2b. Obviously, due to the residual slip boundary condition, the velocities are minimum at the boundaries of the cylinders and that of the middle part are bulgy. For the intermediate narrow cylinder, the velocities are larger than those of the other sections.

In addition, we compute the densities of the flow with external flow field and without external flow field along the *z*-axis, as is shown in Figure 3. Our settled average value of ρ is 3, while the data lines vibrate slightly at about 3, which make sense for the whole system.

### 3.2. Translocation Conformations of the Polymer Chains

Here, we take *N_m_* = 100 for example to illustrate the translocation process. For the trefoil knot chain, there are four situations during the translocation processes, firstly, the knot is unknotted during translocation, which is also the normal state; secondly, the chain has not translocated after all the production steps; thirdly, the chain is super-positioned during the translocation; and lastly, the chain has not unknotted after translocation. Simultaneously, there are three situations for the linear chain—firstly, the chain is translocated normally; secondly, the chain has not translocated after all the production steps; and thirdly, the chain is super-positioned during the translocation. We definitely focus on normal translocation processes, i.e., the first situation for the trefoil knot chain and that of the linear chain. Figure 4a,b illustrates the translocation processes of the trefoil knot chain and the linear chain, respectively, where the stiffness b is 100. For Figure 4a, which shows the translocation process of the linear chain, at *t* = 200*τ*, the linear chain is randomly placed at the left side of the funnel-like channel; through long time relaxation, one end of the chain steps in the narrow cylinder channel at *t* = 11,300*τ*; then the chain translocates the narrow cylinder channel with the external flow. The snapshot at *t* = 15,800*τ* shows the translocation conformation; at *t* = 28,000*τ*, the whole chain completes the translocation, obviously, there is a flow-induced helical coiling in the right side of the channel, which is the same as the work of Raghunath Chelakkot et al. [15]. The transient helix formation is the result of a nonequilibrium and nonstationary buckling transition of the semiflexible polymer, which is subjected to a compressive force originating from the fluid-velocity variation in the channel. For Figure 4b, which shows the translocation process of the trefoil knot chain, at *t* = 300*τ*, the trefoil knot chain is placed at the left wide side of the funnel-like channel; through relaxation, one end of the chain drills into the narrow channel at *t* = 2300*τ*; then the chain passes through the narrow channel under the external flow field. The snapshot at *t* = 4100*τ* shows the translocation conformation; at *t* = 8700*τ*, the whole chain completes the translocation and the knot is untied. There is also a flow-induced helical coiling in the right side of the channel, which is similar to the linear chain. Obviously, the translocation time of the trefoil knot chain is shorter than that of the linear chain, it is because the semi-flexible linear chain needs longer relaxation time to find the entrance, and once it enters into the narrow channel, it may exhibit as a straight line, the driving force of the flow field, which is proportional to the number of squatted monomers at the entrance, will largely decrease; while the trefoil knot chain has more monomers at the entrance because of its knot structure, thus, it has larger driving force from the flow field. One can see its translocation time is shorter.

Both Figure 5 and Figure 6 represent the status once translocation of the trefoil knot chain and the linear chain with various rigidity b complete. Figure 5 shows the comparison of the average gyration radius <Rg> of the linear chain and the trefoil knot chain, versus different rigidity b = 0, 10, 100, 1000 and 10,000, variation with the time steps. The four curves of filled symbols indict the linear chain, and the three curves with hollow symbols represent the trefoil knot chain. Obviously, when b = 0, the curves of the linear chain and that of the trefoil knot are similar to each other, the knot is so small that the effect of the knot on <Rg> is negligible, when the time steps are between 2 × 10^4^ and 8 × 10^4^, there is a peak value of 15, which indicates that the chain is extended during the translocation and no longer a coil; when b = 10, due to the trefoil knot being bigger than that of b = 0 and it cannot be negligible, the profile of the curve is different from that of b = 0, because of it being untied during the whole translocation, its value of <Rg> is larger than that of the linear chain, and the contour of <Rg> is similar to a parabola; while the curve of <Rg> of the linear chain exhibits a sharp peak value of about 15 between 3 × 10^4^ and 5 × 10^4^ time steps. At b = 100 and 1000, the values of <Rg> of the trefoil knot are smaller, between 0 and 4 × 10^4^ time steps, than its linear chain counterpart, afterwards, they are both consistent with each other; for b = 10,000, which implies the chain is a rigid rod, when the trefoil chain cannot translocate the narrow cylinder channel, thus we depict the linear chain here only, obviously, the value of <Rg> at b = 10,000 is about 27.5, it is because the rigid chain maintains the shape of a rod and cannot conduct a shape transformation during the whole process. Obviously, when b = 100 and b = 1000, the contrast between the two counterparts is quite different before translocation, this is because the topological structure of the semi-flexible knot may decrease the contour, while the semi-flexible linear chain can extend freely; while b = 10, the topological structure of knot may inflate the contour; at b = 0, the two counterparts are similar to each other, this is because the chain is a random coil, the knot cannot affect the whole contour.

In order to illustrate the dynamic behavior of the chain in the other aspect, Figure 6 shows the variation of <S> versus time step under the same parameters *N* = 100 and *g* = 0.01. <S> is the mean end-to-end distance of the chain. One can see that the curves of linear chain and trefoil knot chain are similar to each other when b = 0, this is because the trefoil knot chain is a random coil that the influence of the knot on <S> can be negligible, when the time steps are between 2 × 10^4^ and 8 × 10^4^, in the two curves, there are two peak values of about 45, which indicates that the chain is extended and no longer a coil; at b = 10, when the semi-flexible trefoil knot is bigger than its coil counterpart and cannot be negligible, the profile of the <S> curve is similar to a parabola, the whole chain expands due to the knot, the value of <S> is larger than its linear counterpart, whose curve exhibits a sharp peak of about 45 between 3 × 10^4^ and 5 × 10^4^ time steps; when *b* = 100 and 1000, the values of <S> of the trefoil knot chain are smaller than that of the linear chain between 0 and 4 × 10^4^ time steps, this is because the knot topology decreases the contour of the trefoil knot chain; after translocation, the two counterparts are consistent with each other; for b = 10,000, which indicates the rigid chain, and the trefoil chain cannot translocate the narrow cylinder channel during the whole translocation process, thus we show the linear chain here only, obviously, the value of <S> at b = 10,000 is about 96 during the whole translocation process, this is because the rigid chain is maintaining the shape of a rod and cannot conduct a shape transformation during the whole process. Figure 5 and Figure 6 are consistent with each other.

In order to illustrate the influence of the chain length and rigidity on the translocation time, we compute the variation of the translocation time versus the chain length and rigidity. We depict the figures in Figure 7a and Figure 7b, respectively. Figure 7a illustrates the variation of the translocation time step *t* with the chain length *N_m_*; the filled symbols represent the linear chains, and their hollow counterparts indict the trefoil knot chains. Obviously, they both, with different rigidities, have the same trend with the chain length increase. Because of the linear chain forming a false knot before translocation, that is the same thing as tying a chain. One can see this from the snapshots of Figure 4. For the linear chains with different rigidities, the translocation time steps increase with the rigidity increase when b < 100, while the translocation time steps decrease with the rigidity increase when b > 100. This is because the chain is flexible when b = 0, the chain can translocate through the narrow channel easily and has the smallest translocation time steps, while 0 < b ≤ 100, the chain is semi-flexible, it can extend throughout the left cone and wide cylinder, which may delay the chain’s translocation, thus the translocation time increases with the increasing rigidity; however, when b > 100, the translocation time steps decrease with the increasing rigidity. The reason is that, the more rigid the chain is, the more easily the chain is apt to extend to a rigid rod, which can translocate the narrow channel quickly. For the linear chains and the trefoil knot chains with different chain lengths, we get the scale relation of translocation time steps and the chain length $t\sim N_{m}^{1.742}$, which indicates that the translocation time steps increase gradually with the chain length *N_m_*.

Figure 7b illustrates the relation of the chain translocation time steps *t* and the chain rigidity b, the filled symbols represent the linear chains, and their hollow counterparts indict the trefoil knot chains. Obviously, they both, with different chain lengths, have the same trend with the b rigidity increasing. The longer the chain is, the larger the translocation time the chain has. When 0 < b ≤ 100, the translocation time increases with the rigidity monotonously, while the translocation time steps decrease with the increased rigidity when b > 100. This is because the chain is semi-flexible when 0 < b ≤ 100, the chain is semi-flexible, it can extend throughout the left wide channel and the cone channel, which exhibit resistance during the translocation procedure; the more rigid the chain is, the more resistance the chain has, as the rigidity increases, the translocation time steps increase; however, when b > 100, the translocation time steps decrease with the increasing rigidity, the reason is that the more rigid the chain is, the more easily the chain tends to extend to a rigid rod and can translocate through the middle narrow channel quickly due to the lower friction. For the linear chains and the trefoil knot chains with different chain rigidities, we get the scale relation of translocation time steps and the chain rigidity: $t{\sim b}^{0.352}$(when 0 < *b* ≤ 100) and $t{\sim b}^{-0.086}$ (when b > 100), which indicate the relation of the translocation time steps and the chain rigidity b.

## 4. Conclusions

In this paper, we have studied the conformation behaviors of the trefoil knot polymer chain and the linear chain translocate through a funnel-like channel. Firstly, we calculate the characteristic of the Poiseuille flow, the cut—across section of the flow field along the funnel-like channel exhibits parabolic shape and the density along the *z*-axis approaches the settled value, which shows that our simulation is reasonable. Secondly, we calculate the characteristics of the polymer chains, including the trefoil knot chain and the linear chain, including their average gyration radius <Rg> and the average end-to-end distances <S> vary with the time steps, we find the curves are similar to each other when b *<* 100, both of their curves exhibit an ‘n’ shape, while when 100 ≤ b ≤ 1000, they increase first and then tend to be stable. Lastly, we focus on the their translocation time with the chain lengths and rigidities, and we find their scale relations are similar to each other, this is because the linear chain forms a false knot before translocation, this is the same thing as tying a chain; the scale relation of the translocation time versus the chain length is $t\sim N_{m}^{1.742}$, and that of the chain rigidity is $t{\sim b}^{0.352}$ (when 0 < b ≤ 100) and $t{\sim b}^{-0.086}$ (when b > 100).

## Figures and Tables

**Figure 1 polymers-14-01164-f001:**
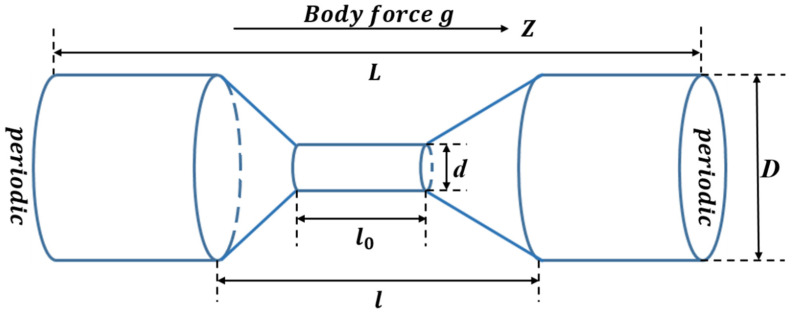
Schematic representation of simulation model for funnel-shape fluidic channel.

**Figure 2 polymers-14-01164-f002:**
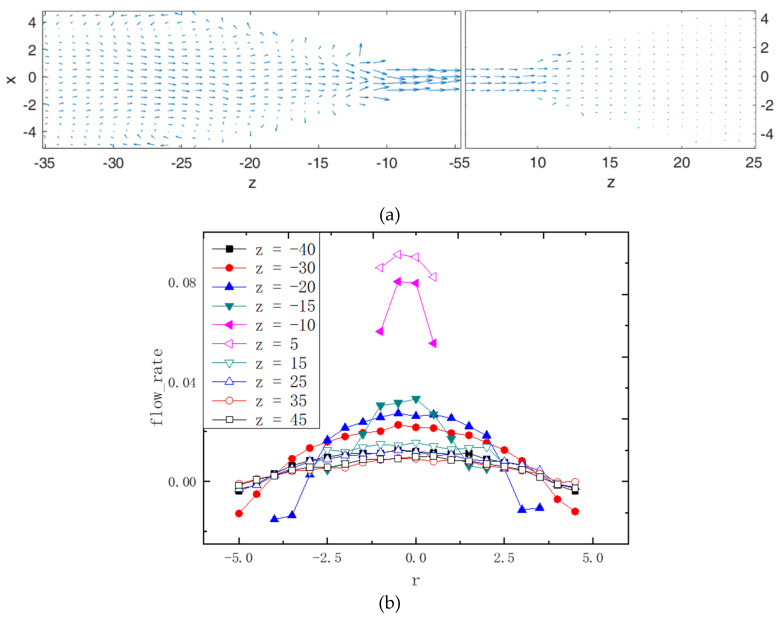
(**a**) 2D vector field of the velocity for the middle plane of the fluidic channel, which indicates cross−section flow field, the arrow direction indicates the flow field direction, the length and the size; (**b**) the lateral velocity along the radius direction with various z, the filled symbols indict the left channel, and the hollow symbols represent the right channel.

**Figure 3 polymers-14-01164-f003:**
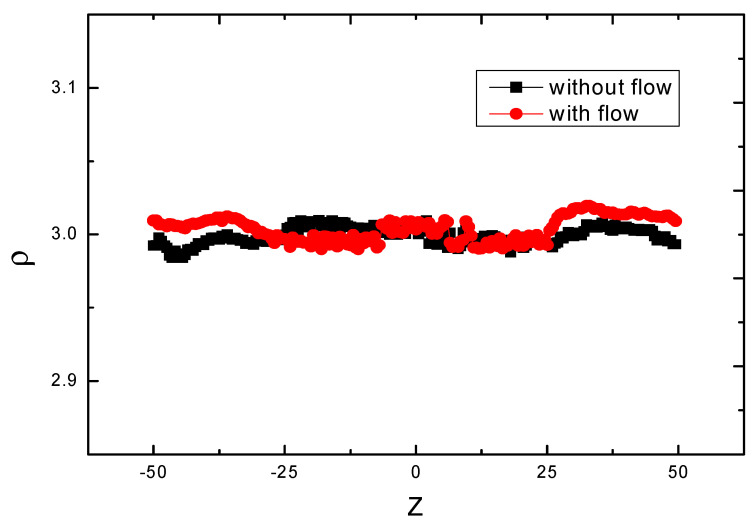
Solvent particle density profiles along the *z*−axis direction of the Poiseuille flow, the square symbols indicate that there is no flow rate added, and the circle symbols represents that there is flow rate added.

**Figure 4 polymers-14-01164-f004:**
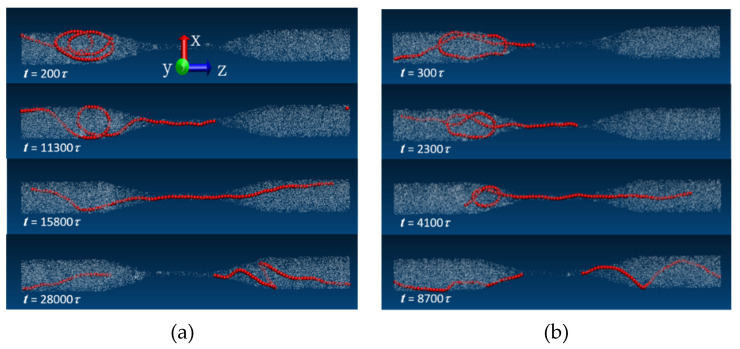
Translocation snapshots of linear chain (**a**) and trefoil knot chain (**b**) with time variation, respectively, where the rigidity b = 100 and the chain length *n* = 100.

**Figure 5 polymers-14-01164-f005:**
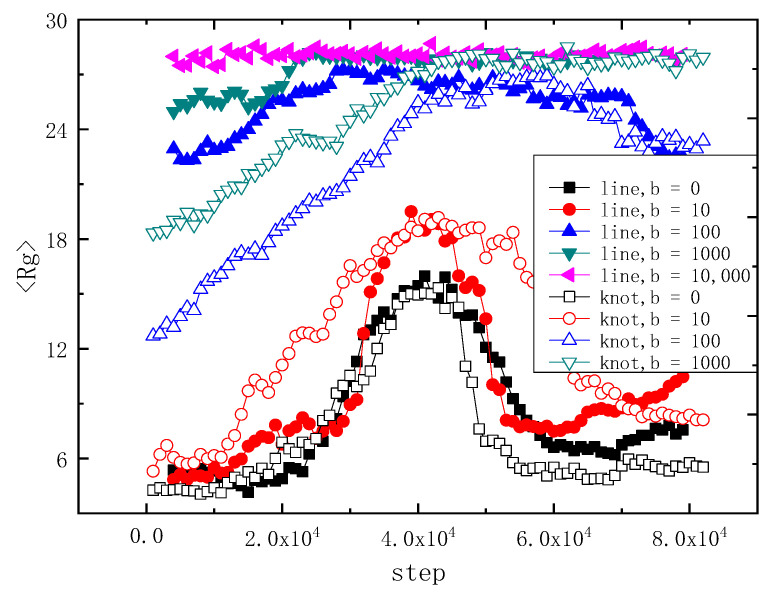
Average gyration radius <Rg> of linear chain and trefoil knot chain, with different bend parameter b, variate with the time steps. There are 100 monomers in each chain. The filled symbols indicate the linear chain, and the hollow symbols represent the 3_1_ knot chain; the same shape and color online symbols represent the same rigidity.

**Figure 6 polymers-14-01164-f006:**
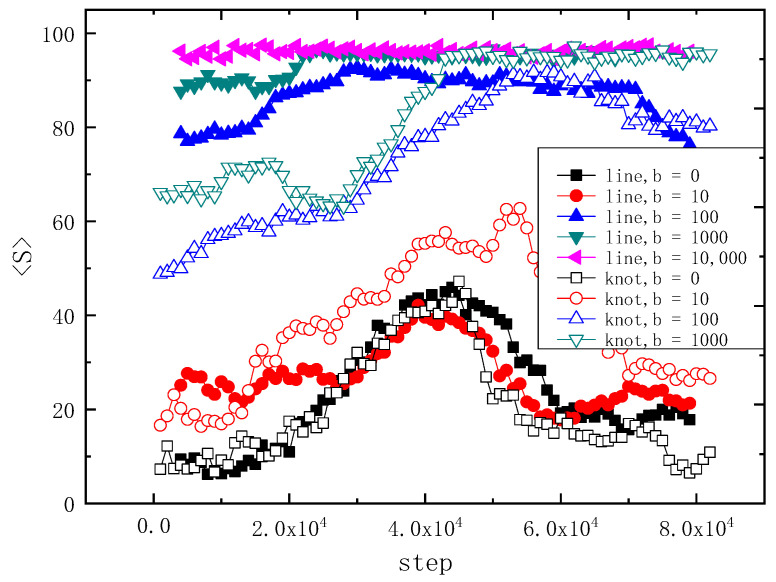
Average end-to-end distances <S> of linear chain and trefoil knot chain, with different bend parameter b, variate with the time step. There are 100 monomers in each polymer chain. The filled symbols indict the linear chain, and the hollow symbols represent the 3_1_ knot chain; the same shape and color online symbols represent the same rigidity.

**Figure 7 polymers-14-01164-f007:**
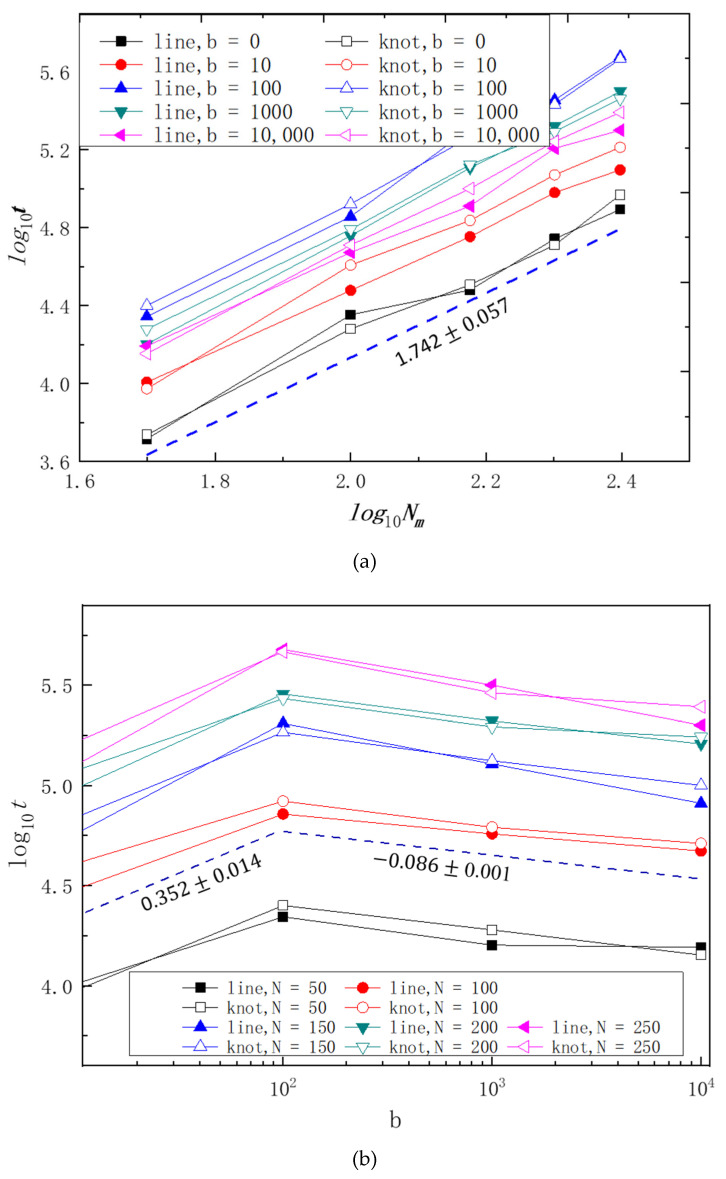
(**a**) Translocation time *t* as a function of polymer chain length *N_m_*. (**b**) Translocation time *t* as a function of bend parameter b. The filled symbols indicate the linear chain, and the hollow symbols represent the 3_1_ knot chain.

## Data Availability

Not applicable.

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
