# Peer review of "A Trefoil Knot Polymer Chain Translocates through a Funnel-like Channel: A Multi-Particle Collision Dynamics Study"

_polymers, 2022, doi:10.3390/polym14061164_

Round 1
Reviewer 1 Report
In this manuscript, the authors study the conformation behaviours of trefoil knot polymer and also linear chain polymer as translocating through a funnel-like channel. This study is interesting, and this reviewer believes it can attract quite a few readers. Herein, this reviewer suggests that this manuscript can be accepted by Polymers after the authors polish the writing and correct some typos.
Author Response
Dear Reviewer :
Thank you for your comment concerning our manuscript entitled “A Trefoil Knot Polymer Chain Translocates through a Funnel-like Channel: A Multi-Particle Collision Dynamics study”. We have studied comment carefully and have made corrections correspondingly. The comment is as follow:
Comment:
In this manuscript, the authors study the conformation behaviours of trefoil knot polymer and also linear chain polymer as translocating through a funnel-like channel. This study is interesting, and this reviewer believes it can attract quite a few readers. Herein, this reviewer suggests that this manuscript can be accepted by Polymers after the authors polish the writing and correct some typos
Reply to comment:
Thanks for your careful checks. Based on your reminder, we tried our best to improve writing and made some changes in the paper, including correct the spelling mistakes and polish the language in the revised manuscript.
Finally, we appreciate for your pertinent suggestion earnestly, and hope that the changes we’ve made resolve your concerns about the article.
Yours sincerely,
Zhiyong Yang
Department of Physics
Jiangxi Agricultural University
Nanchang, China
Reviewer 2 Report
In the present paper, multi-particle collision dynamics for the solvent is combined with molecular dynamics for the polymer in order to analyse the characteristics of a linear polymer chain with trefoil knot in a funnel-like microfluidic device. The influence of confined environment and solvent particles on conformation and dynamic behaviour of the knotted linear polymer chain is investigated.
The paper is well written and organised. Model and simulation approach are correctly introduced. The results of the numerical simulations, which are mainly focused on Poiseuille flow field, conformations and dynamics process of the knotted polymer chain translocation, are properly reported and discussed.
The only aspect to be clarified, in order to consider this work as acceptable for publication, is related to the boundary conditions. Specifically, the Authors are invited to add within the paper, to improve clarity, also the characteristic equations of the boundary conditions adopted, i.e., "confinement" periodic boundary conditions for the solvent and "residual slip" boundary conditions for the channel.
Therefore, by considering the previous notes, in the opinion of the Reviewer the paper should be accepted for publication in Polymers by minor revision.
Author Response
Dear Referee:
Thank you for your comment concerning our manuscript entitled “A Trefoil Knot Polymer Chain Translocates through a Funnel-like Channel: A Multi-Particle Collision Dynamics study”. We have studied comment carefully and have made clarity correspondingly. The comment is as follow:
Comment:
In the present paper, multi-particle collision dynamics for the solvent is combined with molecular dynamics for the polymer in order to analyse the characteristics of a linear polymer chain with trefoil knot in a funnel-like microfluidic device. The influence of confined environment and solvent particles on conformation and dynamic behaviour of the knotted linear polymer chain is investigated.
The paper is well written and organised. Model and simulation approach are correctly introduced. The results of the numerical simulations, which are mainly focused on Poiseuille flow field, conformations and dynamics process of the knotted polymer chain translocation, are properly reported and discussed.
The only aspect to be clarified, in order to consider this work as acceptable for publication, is related to the boundary conditions. Specifically, the Authors are invited to add within the paper, to improve clarity, also the characteristic equations of the boundary conditions adopted, i.e., "confinement" periodic boundary conditions for the solvent and "residual slip" boundary conditions for the channel.
Therefore, by considering the previous notes, in the opinion of the Reviewer the paper should be accepted for publication in Polymers by minor revision.
Reply to comment:
We have specified the issues in the revised manuscript at the last paragraph of section 2.3:
Periodic boundary conditions are applied in the z-direction, while x- and y-directions are confined by cylindrical channel. When a random shift of the cell lattice before streaming is required to guarantee Galilean invariance, partially occupied boundary cells are unavoidable. Then the simple bounce-back rule, i.e. the velocities of particles which hit the wall are inverted after the collision, fails to guarantee no-slip boundary conditions in the case of partially filled cells. To reduce fluid slip at the walls, the virtual particles approach is adopted. That is to say, for all the cells of the channel which are cut by walls and therefore have a number of particles smaller than the average number of the bulk cells, we fill the“wall”part of the cells with virtual particles in order to make the effective density of real plus virtual particles equal the average density. The velocities of the virtual particles are drawn from a Maxwell-Boltzmann distribution of zero average velocity. In one collision step, the relative velocities with respect to center-of-mass velocity of the particles within this cell are rotated, the process is similar to the bulk cells. Specially, the virtual particles are static for simplicity, but their velocities are reset at each step. This boundary condition is called residual slip.
Finally, we appreciate for your pertinent comment earnestly, and hope that the changes we’ve made resolve your concerns about the article.
Yours sincerely,
Zhiyong Yang
Department of Physics
Jiangxi Agricultural University
Nanchang, China